



# Sixteen years of MOPITT satellite data strongly constrain Amazon CO fire emissions

Stijn Naus[1,2], Lucas G. Domingues[3,4], Maarten Krol[1,5], Ingrid T. Luijkx[1], Luciana V. Gatti[4,6], John B. Miller[7], Emanuel Gloor[8], Sourish Basu[9,10], Caio Correia[4,6], Gerbrand Koren[11], Helen M. Worden[12], Johannes Flemming[13], Gabrielle Pétron[7,14], and Wouter Peters[1,15]

[1]Meteorology and Air Quality, Wageningen University and Research, The Netherlands
[2]SRON Netherlands Institute for Space Research, Utrecht, The Netherlands
[3]National Isotope Centre, GNS Science, New Zealand
[4]Nuclear and Energy Research Institute, São Paulo, Brazil
[5]Institute for Marine and Atmospheric Research, Utrecht University, The Netherlands
[6]National Institute for Space Research (INPE), São José dos Campos, Brazil
[7]Global Monitoring Laboratory, National Oceanographic and Atmospheric Administration, Boulder, CO, USA
[8]School of Geography, University of Leeds, Leeds, UK
[9]Earth System Science Interdisciplinary Center, University of Maryland, MD, USA
[10]NASA Goddard Space Flight Center, Greenbelt, MD, USA.
[11]Copernicus Institute of Sustainable Development, Utrecht University, Utrecht, The Netherlands
[12]Atmospheric Chemistry Observations and Modeling, National Center for Atmospheric Research, Boulder, CO, USA
[13]European Centre for Medium-Range Weather Forecasts (ECMWF), Reading, UK
[14]Cooperative Institute for Research in Environmental Sciences, University of Colorado, Boulder, CO, USA
[15]Centre for Isotope Research, University of Groningen, The Netherlands

**Correspondence:** Stijn Naus (s.naus@sron.nl)

**Abstract.** Despite consensus on the overall downward trend in Amazon forest loss in the previous decade, estimates of yearly carbon emissions from deforestation still vary widely. Estimated carbon emissions are currently often based on data from local logging activity reports, changes in remotely sensed biomass as well as remote detection of fire hotspots, and burned area. Here, we use sixteen years of satellite-derived carbon monoxide (CO) columns to constrain fire CO emissions from the Amazon

basin between 2003 and 2018. Through data assimilation, we produce 3-daily maps of fire CO emissions over the Amazon that we verified to be consistent with a long-term monitoring program of aircraft CO profiles over five sites in the Amazon. Our new product independently confirms a long-term decrease of 54% in deforestation-related CO emissions over the study period. Interannual variability is large, with known anomalously dry years showing a more than fourfold increase in basinwide fire emissions. At the level of individual Brazilian states, we find that both soil moisture anomalies and human ignitions

determine fire activity, suggesting that future carbon release from fires depends on drought intensity as much as on continued forest protection. Our study shows that the atmospheric composition perspective on deforestation is a valuable additional monitoring instrument that complements existing bottom-up and remote sensing methods for land-use change. Extension of such a perspective to an operational framework is timely considering the observed increased fire intensity in the Amazon basin in 2019–2021.





## 1 Introduction

The role of Amazon forests in supporting biodiversity, regional ecosystem services, and carbon storage (Gloor et al., 2012) is threatened by human activities, in the form of large-scale deforestation (Davis et al., 2020) and climate change. In Brazil specifically, various studies suggest that, in recent years, deforestation rates and associated fire activity are once again accelerating (INPE, 2020; Pereira et al., 2020), after having reached a minimum around 2012 (Yin et al., 2020). Moreover, recent

droughts in 2010 and 2015/16 led to maxima in biomass burning (Silva Junior et al., 2019). Reliable monitoring of fire activity and its impacts provide objective detection and mapping of deforestation, which helps in investigating underlying drivers. Such information is key for developing efficient mitigation measures and for reducing fire risks.

Monitoring of fires primarily relies on remote sensing products such as fire counts (Wiedinmyer et al., 2011), albedo changes (van der Werf et al., 2017), and Fire Radiative Power (FRP; (Kaiser et al., 2012)), which are partly related (Fanin and van der

Werf, 2015). Rapid and continuous processing of vast amounts of such data allowed recent unexpectedly high fire activity in 2019 to be detected and reported quickly (Lizundia-Loiola et al., 2020; Brando et al., 2020). However, fire dynamics are complex, and products based on land remote sensing data are prone to miss small fires (Randerson et al., 2012; Ramo et al., 2021), hampered by cloud cover (Schroeder et al., 2008), and might be poorly able to detect understory fires (Morton et al., 2013). Understory fires in particular contribute strongly to forest fragmentation and mortality, and can increase forest vulnerability

to burning (Nepstad et al., 2001; Alencar et al., 2004). In addition to direct detection of fire activity, information is needed to quantify the corresponding carbon loss to the atmosphere on scales from decades to seasons, and from the entire Amazon basin down to individual Brazilian states.

During fires, a combination of pollutants is released into the atmosphere, the composition of which depends on local fire conditions, but which generally includes a large contribution from carbon monoxide (CO). With an atmospheric lifetime of

1–3 months, CO is not well-mixed globally, so that fire emissions produce large and thus easily detectable enhancements over the CO background concentration. Therefore, enhancements in CO over and around the Amazon basin can inform on the frequency, intensity and location of fires. Moreover, the CO released from fires that escaped direct detection such as under cloud cover, or in the understory, can still be detected. Finally, CO fire emissions can be linked to total carbon emissions and emissions of other pollutants with the use of emission factors (e.g. Ferek et al. (1998); van Leeuwen et al. (2013)), giving

insights in climate and air quality impacts of fires.

Satellite data of CO are especially useful for quantifying and mapping fire emissions, due to their temporal and spatial detail, and their availability in remote areas. In this work, we focus on the use of satellite-detected CO column retrievals from the Measurement Of Pollution In The Troposphere (MOPITT) instrument (Deeter et al., 2019), which is an established product for CO emission quantification (Jiang et al., 2017; Miyazaki et al., 2020). For the Amazon basin specifically, MOPITT CO data

were analysed previously to show that a long-term decrease over 2002–2016 in deforestation is partly counteracted by large fires in drought years (Aragão et al., 2018; Deeter et al., 2018).

Here, we move beyond direct analysis of satellite data, and incorporate these data in the data assimilation system TM5-4DVAR (Krol et al., 2005; Meirink et al., 2008). By linking satellite data to the 3D transport model TM5, we can map and



quantify CO fire emissions in the Amazon with improved detail and accuracy. MOPITT CO satellite data were previously
analysed by Zheng et al. (2019) in a comparable modeling framework to assess the CO budget from 2000 to 2017 on a global
scale, and with a focus on fire emissions in Yin et al. (2020). Other data assimilation studies that were focused on South-
America have provided insight into fire and drought events, but generally covered shorter time-periods (Hooghiemstra et al.,
2012; van der Laan-Luijkx et al., 2015).

We first present fire emissions over the entire Amazon basin (Section 3.1.1), and then zoom in on individual Brazilian
states (Section 3.1.2), and on different landcover types (Section 3.1.3). This level of detail helps in better understanding and
quantifying differences between bottom-up and top-down estimates, and in assessing anthropogenic and natural contributions
to fire emissions. An important asset of our analysis is a detailed investigation of the uncertainties in the Bayesian inverse
system. To this end we investigate the influence of components such as the prior fire CO emission inventory, natural production,
loss fields, and we additionally assimilate a different satellite product. Uniquely, we independently assess our MOPITT-based
emission estimates, as well as those from the GFAS bottom-up inventory, using a multi-year CO record from an aircraft whole
air flask sampling network in the basin (Gatti et al., 2014, 2021) (Section 3.2.2).

## 2 Methods

### 2.1 Transport model

We operate the atmospheric transport model TM5 (Krol et al., 2005) at a global resolution of 6° longitude by 4° latitude.
We additionally use the zoom capability of the TM5 model to include two nested regions over South-America, in a set-up
similar to van der Laan-Luijkx et al. (2015) (see Fig. 1). The inner zoom domain (red in Fig. 1), with a resolution of 1° by
1°, spans longitudes from 75°W to 39°W and latitudes from 28°S to 8°N. The outer zoom domain (green in Fig. 1), with a
resolution of 3° longitude by 2° latitude, spans longitudes from 84°W to 30°W and latitudes from 34°S to 14°N. Note that we
present most results for the inner zoom domain only. All regions are operated with 25 vertical layers, covering the range from
surface pressure to top of the atmosphere, with typically 3–6 model layers in the planetary boundary layer. Transport in TM5
is driven by 3-hourly, offline meteorological fields from the ERA-Interim reanalysis from the European Centre for Medium
Range Weather Forecasts (ECMWF) (Dee et al., 2011).

### 2.2 Prior source and sink fields

We performed 2003–18 inversions with three prior fire inventories: the Global Fire Assimilation System (GFASv1.2; Kaiser
et al. (2012)), the Fire INventory from NCAR (FINNv1.5; Wiedinmyer et al. (2011)) and a climatological (i.e. annually repeat-
ing) prior based on the average emission distribution in GFAS (hereafter referred to as CLIM). The GFAS emission distribution
is provided on 0.5° by 0.5° resolution, and the FINN distribution is available at ~ 1 km by 1 km resolution, but we regrid both
to our model resolution. The GFAS and FINN inventory use different data from the Moderate Resolution Imaging (MODIS)
satellite as a proxy for the global, daily fire distribution: for GFAS fire radiative power (FRP); for FINN active fire counts.





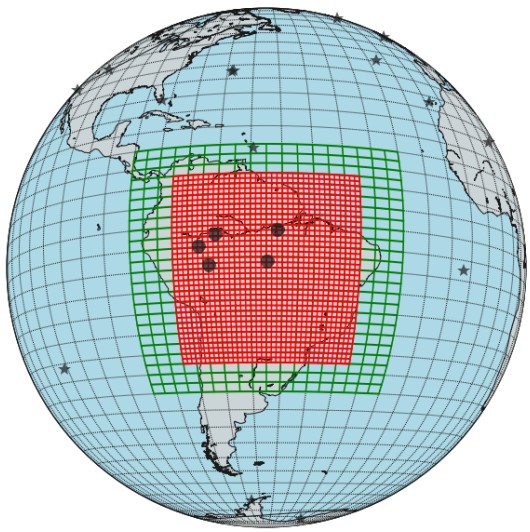

**Figure 1.** A map of the South-American zoom regions used in the TM5 simulations. The green and red grids indicate the $3°$ longitude by $2°$ latitude and $1°$ longitude by $1°$ latitude zoom regions, respectively. Over both zoom regions satellite data are assimilated. Filled stars indicate NOAA Global Greenhouse Gas Reference Network surface sites, from which surface observations are assimilated to constrain global CO emissions, and by extent the boundary conditions of the zoom domains. Filled circles indicate sites where discrete whole air samples were collected by aircraft at various altitudes. The whole air sample calibrated CO data are used for independent evaluation of the inversions.

80  Both inventories overlay these proxies with landcover maps from the MODIS instrument, and use landcover-specific emission factors for CO (and other species) to produce the emission estimates we use. However, the exact landcover classifications and emission factors used differ between the two inventories. We use the inversions that start from the GFAS prior as reference, and discuss the differences between the fire priors where relevant.

Fire emissions retrieved in our inverse system are informed both by the prior emission distribution (e.g. GFAS) and by the
85  assimilated observational data (e.g. MOPITT). To assess the importance of either component we have constructed a climatological prior (CLIM), which includes no interannual variability and no spatial gradients. As such, variability in the posterior emissions retrieved in the CLIM inversions is driven exclusively by the assimilated CO data and atmospheric transport. Since we optimize CO fire emissions at three-day resolution (Section 2.6), we also construct the CLIM prior at three-daily resolution. We construct the CLIM prior as follows. First, we average the daily CO emission fields from GFAS over the 2003–2018 period.
90  Next, if in a three-day period the total CO emissions in a grid cell are less than 0.03 Tg, the emissions in that grid cell and three-day window are set to zero. Finally, three-daily total fire emissions inside the $1°$ by $1°$ zoom domain are divided uniformly over those grid cells that initially had emissions higher than 0.03 Tg. In this way, approximately 24% of the inner domain grid cells





still contain emissions. We choose this approach to prevent spatial gradients in the GFAS estimate from influencing posterior emissions, which is balanced by the 0.03 Tg lower limit to retain some potential to recover high, localized emissions from the CLIM prior. Outside the $1° \times 1°$ zoom domain, the CLIM prior still includes interannually varying GFAS emissions.

All fire emissions (GFAS, FINN and CLIM) are distributed vertically in the simulations following vertical emission profiles derived in the Integrated System for wild-land Fires (IS4FIRES) (Soares et al., 2015). Different from the horizontal emission distribution, the vertical emission distribution does include diurnal variability. In a sensitivity test we have found that including a diurnal cycle in only the vertical emission distribution produces results that are comparable to including diurnal variability in both the horizontal and vertical distribution of CO fire emissions (results not shown). Therefore, we do not impose sub-daily variability on the native daily resolution of GFAS and FINN emissions in the simulations.

For chemical production of CO from methane ($CH_4$) and from non-methane volatile organic compounds (NMVOCs), we use fields generated in a 2006 simulation in the full-chemistry version of TM5 (Huijnen et al., 2010). We use anthropogenic CO emissions from the Monitoring Atmospheric Composition and Climate Cityzen (MACCity) inventory (Lamarque et al., 2010). The annually repeating, monthly hydroxyl radical (OH) concentration fields are a combination of tropospheric OH fields from (Spivakovsky et al., 2000), scaled by 0.92 as recommended in (Patra et al., 2011), and stratospheric OH fields derived in the 2D Max Planck Institute for Chemistry (MPIC) chemistry model (Brühl and Crutzen, 1993). In Sect. 3.3.3, we discuss the sensitivity of the inversion results to the chemical CO production and to the OH distribution.

### 2.3 Satellite retrievals

In the reference inversions, we assimilate CO column retrievals from the Measurements Of Pollution In The Troposphere (MO-PITT) instrument, version 8 (Deeter et al., 2019) over both South-American zoom domains. MOPITT retrieves CO columns using both the CO absorption band in thermal infrared (TIR) at 4.7 μm, and in near infrared (NIR) at 2.3 μm. In this work, we only use CO columns retrieved in the TIR band. MOPITT has a swath of 22 by 650 km, with 116 cross-swath pixels, with a daily overpass between 1.00 and 3.30 local time each afternoon and night for the inner zoom domain. Following Nechita-Banda et al. (2018), we inflate column errors reported by the MOPITT team with a factor $\sqrt{50}$ to compensate for the high number of satellite data ($\sim 10.000$/day in both zoom domains combined), relative to the number of surface observations ($\sim 150$/month; see also Sect. 2.4). CO columns are sampled from the transport model using the MOPITT averaging kernels.

In Sect. 3.3.1, we present inversions in which satellite data from the Infrared Atmospheric Sounding Interferometer (IASI) instrument (Clerbaux et al., 2009) are assimilated instead of MOPITT. IASI retrieves CO columns exclusively in the TIR waveband, with a $12 \times 4$ km footprint at nadir. Identical versions of the IASI instrument fly on-board of three operating platforms (Metop-A, -B and -C), but in this work we limit ourselves to the Metop-A data, which covers the longest time period. Importantly, while IASI and MOPITT exploit similar wavebands, they use different measurement techniques (George et al., 2015). The overpass time of IASI typically precedes the MOPITT overpass time by one hour.





## 2.4 Surface observations

To constrain global emissions of CO outside of the model zoom domains, we assimilate CO mole fraction observations from the surface whole air flask sampling network (40–45 sites) of the NOAA Global Greenhouse Gas Reference Network (GGGRN) (Petron et al., 2019). We use a fixed observational error of 2 ppb CO per flask pair average, with no model error. We choose not to adopt a model error for surface observations, since this would only require further inflation of the error in satellite data (see above). We test the effect of reducing this observational error to 0.2 ppb CO in Supp. 1.

## 2.5 Aircraft observations

We use a set of aircraft observations over the Amazon for independent validation of the inverse results over the 2010–2017 period. Atmospheric air samples were collected at a range of altitudes over five sites in the Brazilian Amazon (Gatti et al., 2014, 2021). The Tabatinga site (TAB) was replaced in 2013 by Tefé (TEF), and we generally group these two sites in our analysis. Site locations are indicated in Fig. 1 and Fig. 6. The sampling flights were performed using small aircraft, typically

two times per month, between 12–13 hr local time, in a descending helicoidal profile that avoids sampling emissions from the aircraft. One profile typically includes 12–17 air samples between 300 m and 4500 m height above sea level (Gatti et al., 2014, 2010). The concentrations of the aircraft samples are analyzed since 2015 at INPE (São José dos Campos, Brazil) - LaGEE (Greenhouse gas laboratory) and before this time at IPEN in an Atmospheric Chemistry Laboratory in São Paulo. The LaGEE uses standards calibrated against the World Meteorological Organization reference scales maintained by NOAA

Global Monitoring Laboratory (i.e. the same calibration scale that is used for NOAA GGGRN surface observations). In the period during and after this transition (2015–16), operation of the aircraft network was partly interrupted (see also the right-most column in Fig. 6). A 2010–2013 subset of these aircraft data was previously used for direct validation of MOPITT CO data (Deeter et al., 2016). However, since we sample both satellite and aircraft data from a 3D-simulated atmosphere, our validation accounts more realistically for the different vertical sensitivities of the two datasets.

## 2.6 Optimization procedure

We employ the TM5-4DVAR inverse system (Meirink et al., 2008) to optimize CO emissions between 2003 and 2018 in 16 separate inversions, which each cover the April–December period of 1 year (i.e. centered on the Amazon fire season). Simulating 9 instead of 12 months greatly improves the speed of convergence of the inversions, since the complexity of the inverse problem scales non-linearly with the number of assimilated observations and optimized state elements. Since Amazon

fires occur mostly between June and November, this set-up still provides us with a one-month spin-up and spin-down period.

The CO satellite data are only assimilated inside both zoom domains over South-America. In the zoom domains, we optimize three-day total CO fire emissions, with a relative grid-box error on the emissions of 250%, a horizontal correlation length of 200 km and a temporal correlation of three days. Total emissions in the global domain are constrained mainly from NOAA surface observations, and are optimized with a prior uncertainty of 250%, and with a horizontal and temporal correlation

of 1000 km and 9.5 months, respectively. Emissions are optimized non-linearly, following a semi-lognormal distribution, to





prevent negative posterior emissions, as in Bergamaschi et al. (2009). Since the 4DVAR system does not produce posterior error covariance matrices for a non-linear system, we instead explore the uncertainties of the inverse system in sensitivity tests (e.g. adjusting the prior emission distribution, adjusting the observational errors and assimilating a different satellite product; see Sect. 3.3 and Supp. 1 and 2).





## 3 Results

### 3.1 Flux analysis

#### 3.1.1 Basin-wide Amazon fire emissions

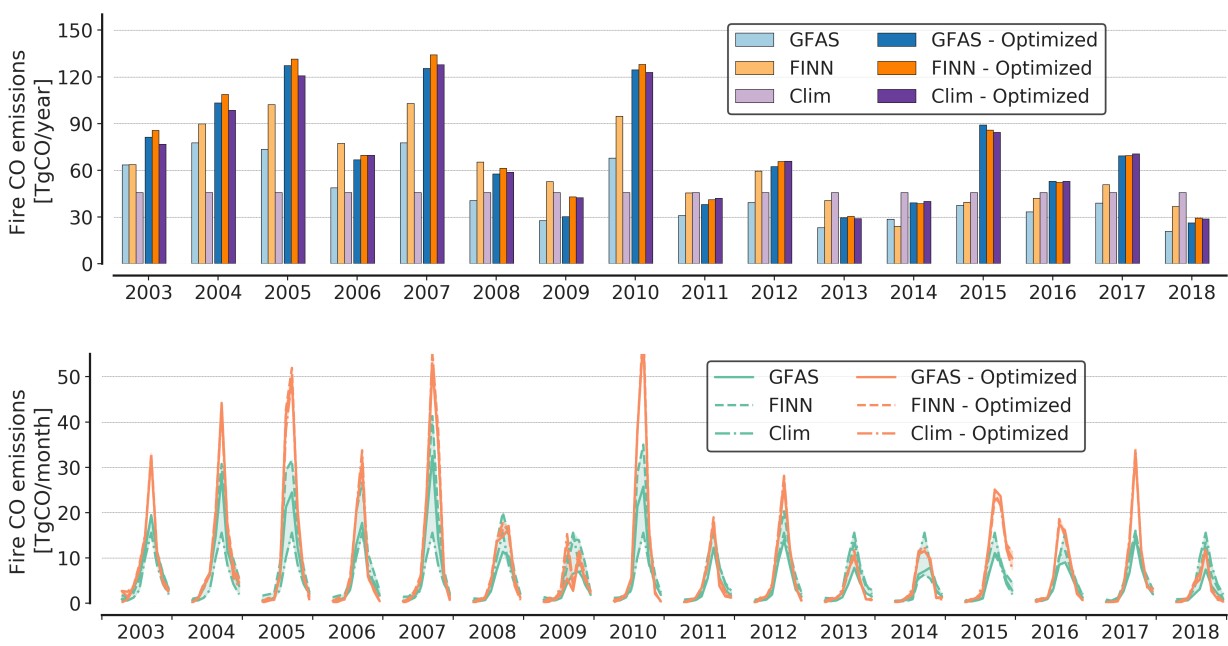

**Figure 2.** Prior and posterior fire CO emission estimates summed over the 1° by 1°, South-American zoom domain (see Fig. 1). Top panel: May–Dec total emissions per year. Bottom panel: Monthly total emissions, with colors distinguishing prior (green) from posterior (orange), and line-style indicating the prior used. Shaded areas mark the spread between the three priors and the three posteriors respectively. Note that the posterior estimates mostly overlap, so that these are visible as one line only.

The inverse system retrieves annual total (May–Dec) fire emissions that vary strongly interannually (top panel in Fig. 2). Typically, GFAS, FINN and the posterior estimates show highest and lowest emissions in the same years. However, the posterior interannual variability is significantly larger than the prior variability, and especially in high-emission years prior inventories underestimate the emissions that are needed to match the MOPITT CO data.

The spread between posterior estimates is significantly lower than the spread between prior estimates. This is an especially noteworthy result for the inversions based on the CLIM prior, which does not include any interannual variability. This result shows that the posterior variability is almost exclusively driven by MOPITT data and TM5 transport, rather than by prior information from the fire inventories. Therefore, we consider that the overall agreement between the prior and posterior interannual variability shows that the fire inventories are generally well able to identify high-emission years.





The fire emissions show a strong seasonal cycle (bottom panel in Fig. 2), with low emissions in April–May and December in most years, in both the prior and posterior emission estimate. This finding generally supports the use of a 9-month inversion period, and confirms that treating April and December as spin-up and spin-down months, respectively, does not strongly affect
the annual total CO fire emissions.

In this 16-year record, 2015 is the year that breaks from these general conclusions. Firstly, while it does not show up as a high-emission year in GFAS and FINN, it does show up as a high-emission year in all posterior estimates. Additionally, it is the only year with high emissions in November and December. Therefore, it is possible that the 2015 estimate is affected by spin-down effects and that we miss emissions in early 2016. However, in an inversion from Nov 2015 to May 2016, we
find that the Jan 2016 emissions are low, and that Dec 2015 emissions are not significantly affected by extending the inversion window (results not shown).

### 3.1.2 Fire emissions in Brazilian states

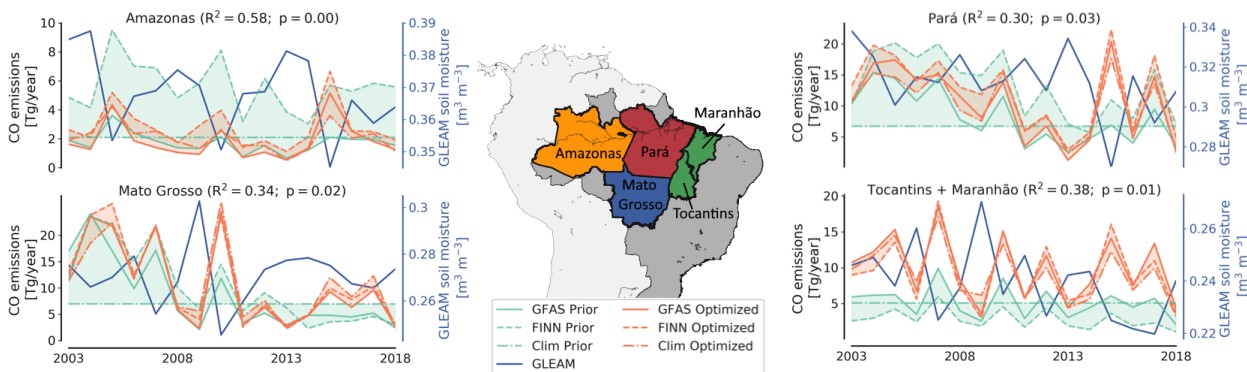

**Figure 3.** Interannual variability in area-average soil moisture and fire CO emissions in five Brazilian states. The four outside panels show the 2003–18 timeseries of May–December total CO emissions from biomass burning (left axis, in orange and green) and annual averaged root-zone soil moisture from the GLEAM v3.3a product (right axis, in blue), for five Brazilian states. A year is defined here as June–December, i.e. centered on the dry season. Both prior (green) and MOPITT-optimized (orange) fire CO emissions are shown, for the three inversions that start from different prior fire inventories (indicated by solid, dashed and dash-dotted lines). Shaded areas indicate the spread between the fire inventories before (green) and after (orange) the inversion. Also shown at the top are the correlation coefficients $R^2$ between CO emissions and soil moisture for the optimized emissions from the inversion that start from the GFAS inventory. Maranhão and Tocantins are combined, since they are smaller states that represent similar regimes in terms of climate and anthropogenic activity. Note the different y-scales between the four figures, with lowest emissions in Amazonas. The centre figure shows a map of these five Brazilian states.

We have quantified interannual variability in fire CO emissions for five Brazilian states to assess these emissions at a sub-basin scale (green and orange lines in Fig. 3). As an indicator of relative drought between years, we have also shown the local
root-zone soil moisture anomalies, as provided by the Global Land Evaporation Amsterdam Model (GLEAM v3a (Gonza-lez Miralles et al., 2011; Martens et al., 2017); blue lines and right axes in Fig. 3). Mato Grosso and Pará make up most of





the Brazilian Arc of Deforestation, and Maranhão and Tocantins (grouped in our analysis) are on the edge of it. Amazonas represents a more pristine area in the Amazon basin. Together, these five states cover most of the Brazilian Amazon.

We find a strong link between interannual variability in optimized CO fire emissions and local soil moisture anomalies. Interannual variations in CO emissions are markedly different per state and, in general, years with low soil moisture levels show high fire emissions. For example, the 2010 fires are mostly located in Mato Grosso, while the 2015/16 fires are concentrated in Pará and Maranhão/Tocantins, which coincides with local, negative anomalies in soil moisture. The GFAS and FINN prior emissions are similarly anti-correlated with soil moisture. However, we also find a similar correlation for the CLIM inversions (result not shown), which does not include prior interannual variability. This result confirms that our inverse set-up can retrieve state-level interannual variability independent from prior assumptions. Moreover, especially in 2015, emissions are strongly adjusted in the inversion, and this coincides with local negative soil moisture anomalies.

We have quantified the correlation between the optimized GFAS emissions and soil moisture anomalies, and we find that it is significant for all states, at significance level p=0.05 (correlation coefficients per state are shown in Fig. 3). The correlation coefficients are largely insensitive to the averaging approach for soil moisture (e.g. selecting the annual minimum value, or averaging over fewer months). Previous work had already established a strong link between fire emissions and soil moisture (e.g. Asner and Alencar, 2010; Silva et al., 2018), but the strength and consistency of the anti-correlation we find here, at the level of individual states, is noteworthy. Of course, drought is not the only determinant of fire emissions, and in Section 3.1.3 we explore the role of deforestation.

Similar to the basin-wide emissions (Fig. 2), we find that spread between posterior estimates at state-level is much reduced compared to the spread between prior estimates. The posterior estimates do not only agree in their interannual variability, but also in the spatial allocation of the emissions between states. For example, the GFAS emissions in Amazonas are much lower than the FINN emissions, yet our inverse results converge towards the lower GFAS estimate. This shows that not only the Amazon-total fire CO emissions, but also their spatial allocation are well-constrained by the inversions.

### 3.1.3 Longterm trends and landcover type

We observe a significant downward trend in the MOPITT-derived CO emissions over our full inner zoom domain (Fig. 4b), which is largely insensitive to the prior inventory used in the inversion. We focus here on five-year averaged emissions, to visualize the clear downward trend that is otherwise partly masked by interannual variations in emissions (e.g. Fig. 3). We disaggregate the trend between forests and savanna, based on annual landcover data from from the Moderate Resolution Imaging Spectroradiometer (MODIS) Land Cover Climate Modeling Grid (CMG), Version 6 (Sulla-Menashe et al., 2019) (Fig. 4). We find that the decrease over the study period (between the 2003–2007 and the 2014–2018 averaged fire CO emissions) is larger over forest ($\sim 54\%$) than over savanna and shrublands ($\sim 39\%$), and the stronger trend over forests is matched in magnitude and sign by a decrease in the independently derived estimates of deforestation from the Brazilian Amazon Deforestation Monitoring Program (PRODES; (INPE, 2020)). The PRODES deforestation rates quantify the area deforested each year, that had not been deforested before. Notably, both deforestation rates and fire emissions have stopped decreasing since $\sim$2012. Deforestation, and especially the narrow definition used in PRODES, does not always overlap with fires, for example when




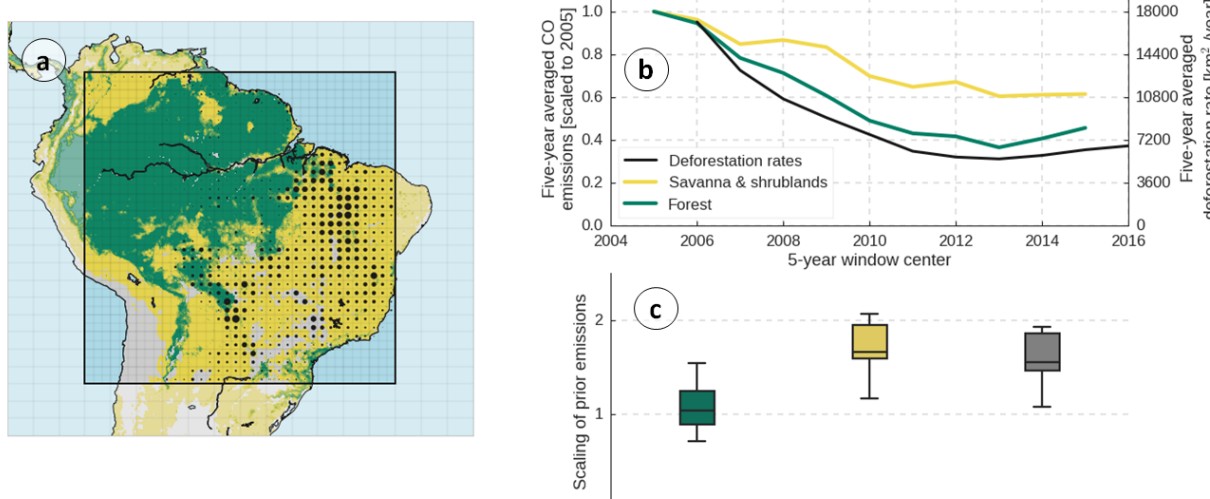

**Figure 4.** Spatial allocation of emissions between landcover types, and longterm trends in the emissions. **Panel a**: A land-cover map of South-America, with landcover types retrieved from the MODIS product, at 0.1° by 0.1° resolution. We distinguish between forest (green), and savanna and shrublands (yellow). Grid cells covered by less than 70% of both are shown in gray. The map covers the 3° by 2° zoom region used in TM5, and the black outline indicates the 1° by 1° zoom region, which is the focus of our study. The area of the black circles is proportional to how much CO is added in the inversion to the GFAS prior emissions, over the 2003–2018 study period. **Panel b**: Timeseries of five-year averaged annual total CO emissions from biomass burning over the 1° by 1° TM5 zoom domain for two landcover types: forest (in green) and savanna/shrublands (in yellow). Emission totals are scaled to the emissions in the first five-year window (see left y-axis), which is centered on 2005. Also shown are five-year averaged deforestation totals (black, right axis), as retrieved from the the Brazilian Amazon Deforestation Monitoring Program (PRODES). Note that the area used for PRODES deforestation rates is different from the forest-mask we use for emission attribution, as PRODES, among other differences, only quantifies deforestation in Brazil. **Panel c**: Box-plot that shows how much annual total emissions over each landcover type are adjusted through optimization with MOPITT satellite data. This can be compared to the black circles in Panel a.

fires occur in reforested areas, or when cut forest is burned with delay or not at all. However, most fires are in some way caused by local anthropogenic activity, for which deforestation is a good proxy. The close match in trends shows that fire abatement policies do reduce fire emissions, but this is often masked by interannual, drought-driven variations. This conclusion confirms similar assessments in earlier MOPITT-based work (Aragão et al., 2018; Deeter et al., 2018), but here, with the use of the

TM5-4DVAR inverse system, we are able to quantify both the decrease and drought-driven interannual variations.

The prior CO emissions from both GFAS and FINN are too low to reproduce MOPITT CO column retrievals in all years, for all landcover types, but the underestimate is much stronger over savanna and shrublands than over forests (Fig. 4c and black circles in Fig. 4a). A strong systematic underestimate over savanna/shrublands is indicative of underestimated CO emission factors and carbon stock, since other explanations, such as missed small fires or understory fires, are more likely to impact

emission estimates from forests. We do note that the amplitude of the underestimate over savanna/shrublands (median 67%) is





large compared to typical uncertainties in emission factors (e.g., van Leeuwen et al. (2013)) and carbon stock. As noted earlier, we find that emissions in dry years, such as 2010 and 2015, are underestimated more strongly than emissions in wet years. We find that the FINN and GFAS inventories underestimate fire emissions most strongly in 2015, which could in part be driven by the timing of these fires. The 2015 fires continued into November and December (see Section 3.1.1): months that typically

have more cloud-cover, which inhibits direct fire detection.

## 3.2 Comparison to observations

In this section, we assess the skill of our prior and posterior simulations to reproduce the assimilated satellite data, as well as independent aircraft profiles of CO. A comparison with surface observations is presented in Supp. 1. Results presented in this section are largely insensitive to the prior fire inventory used, which is why we only present results from the reference GFAS

inversions.

### 3.2.1 MOPITT satellite retrieval

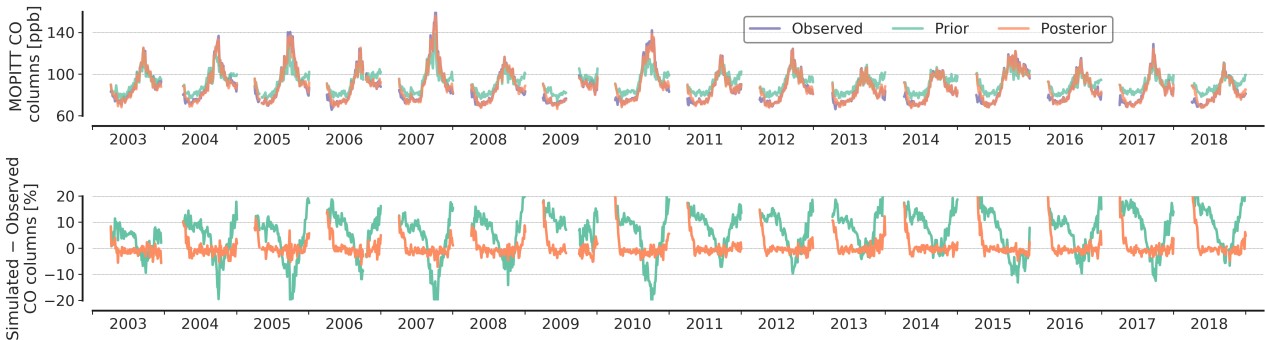

**Figure 5.** A comparison between simulated and MOPITT-retrieved CO columns for the GFAS reference inversion. Top panel: TM5-simulated and MOPITT-retrieved CO columns, averaged over the inner 1° by 1° zoom domain and at three-daily time resolution. Note that the posterior simulation and the MOPITT-retrieved CO columns largely overlap, such that the latter is poorly visible. Bottom panel: The relative difference between satellite-retrieved and simulated MOPITT CO columns, in a simulation with the prior GFAS emissions (green) and after the MOPITT CO column retrievals have been assimilated to optimize fire emissions (orange), averaged as in the top panel. The difference between simulated and observed columns is quantified as a percentage of the average satellite-retrieved CO column.

The MOPITT-retrieved CO columns show a distinct seasonal cycle that in most years peaks in September (Fig. 5, top panel), similar to the fire emissions (Fig. 2). In the prior simulations, we find significant differences between the domain-averaged simulated and satellite-retrieved CO columns (Fig. 5, bottom panel). In the posterior, these differences are reduced to less

than 2% of the observed columns. The nine-month inversion window does show a spin-up period of approximately one month in which the difference is larger, since we do not optimize the initial CO distribution. The posterior agreement between the simulations and the MOPITT data confirms that with adjustments in biomass burning emissions inside the zoom domains and





with adjustments in total emissions outside, we can reproduce satellite-retrieved MOPITT CO columns in our simulations within their observational errors.

Differences between simulated and observed columns in the prior simulation are indicative of the types of adjustments in CO emissions that are needed to reproduce observed columns. Firstly, simulated CO columns outside the dry season are systematically too high, by about 8-15%. Secondly, superimposed on this systematic overestimate, we find that the peaks in CO column retrievals during dry season are underestimated in the prior simulations. The overestimate we attribute largely to too-high secondary production of CO (further discussed in Supp. 2.1). Since we do not optimize secondary production in our

inversion, the overestimate is largely corrected by adjusting emissions outside the inner zoom domain, which are only loosely determined by the NOAA surface observations (see Sup. 1). The prior underestimate of the dry-season peak in CO is most likely related to fire emissions, and we indeed find that fire emissions are increased after data assimilation (Section 3.1.3).

### 3.2.2    Validation with aircraft profiles

Whole air flask sampling flights were conducted over five sites in the Amazon basin between 2010 and 2017 (Gatti et al.,

2021). We compare the TM5-simulated CO mole fractions to those measured from the samples of vertical profiles (Fig. 6). This independent validation clearly shows an improved overall match after assimilation of the MOPITT-retrieved CO columns. In simulations with prior GFAS emissions we find a site-averaged bias of –62 ppb, which reduces to –19 ppb after the optimization (visible in left columns of Fig. 6). This improvement confirms that the prior GFAS emissions are too low to reproduce observations. For Santarém the residual bias is largest (–32 ppb CO after optimization), but MOPITT CO column retrievals

over the same region are matched well after optimization (Supp. 3). We do find a significant absolute residual error between simulated and observed aircraft profiles, which can be explained by the relatively coarse resolution of the transport model (1° by 1° with 25 vertical layers; see Methods), which puts a limit on how well individual aircraft samples can be represented in TM5. However, we find that, in general, the MOPITT-derived emissions much improve the agreement with independent aircraft profiles at five different locations across the Amazon, compared to the GFAS prior. This gives us confidence that our inver-

sion improves estimates of CO emissions from fires across different regions of the Amazon Basin, for example for different Brazilian states (Section 3.1.2).

### 3.3    Sensitivity tests

### 3.3.1    Assimilating IASI instead of MOPITT satellite data

We have performed additional inversions in which we assimilate CO column retrievals from the IASI-MetopA instrument

(Clerbaux et al., 2009). We find that before 2014, MOPITT-derived emissions are slightly higher than IASI-derived emissions (6–12 Tg CO/year), while after 2014 IASI-derived emissions are significantly higher than MOPITT-derived emissions (15–30 Tg/year). Within each of these two time periods the difference between interannual variability in IASI- and MOPITT-derived emissions is small (∼10 Tg/year) relative to this jump (∼30 Tg/year).





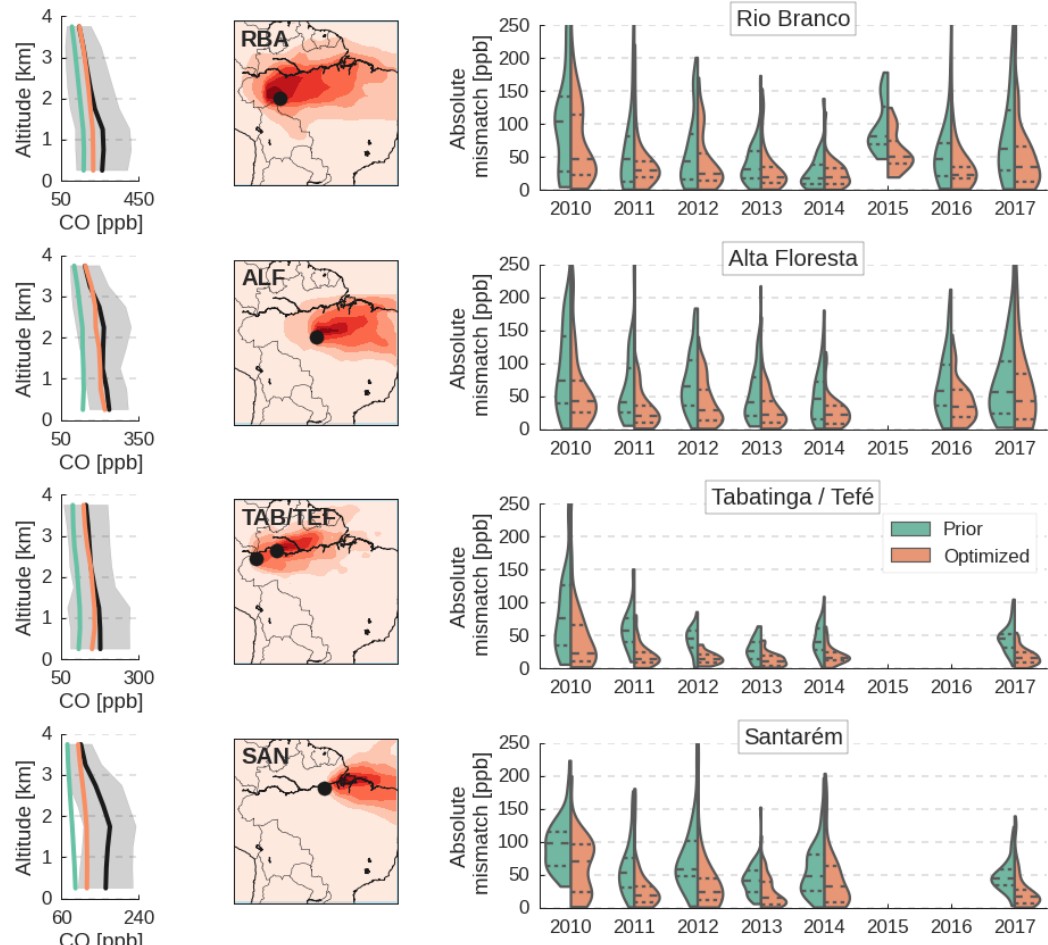

**Figure 6.** Comparison between simulated and observed aircraft profiles over five sites in the Amazon. Profiles sampled over Tabatinga and Tefé are combined, since they represent similar air masses and have complementary temporal coverage (Tabatinga up to 2013; Tefé after 2013). The profiles cover the 2010-17 period, and we have included only profiles sampled between August–November. **Left column:** Time-averaged (2010–2017) simulated (prior in green; posterior in orange) and observed (black) aircraft profiles, binned in 500 meter intervals. Grey shaded areas show one standard deviation of the variability between the individual, observed aircraft profiles. **Center column:** Maps of the influenced area of each site, or site combination. Black dots indicate site locations, and the red area indicates the origin of air at the site location. Red areas are proportional to the logarithm of the number of back-trajectories that originate at the sampling location and altitude, and then pass through a grid cell, as determined from simulations in the HYSPLIT model. Further details are provided in (Gatti et al., 2010). Backtrajectories from the Lagrangian grid in the HYSPLIT model were interpolated to the TM5 1° by 1° grid. **Right column**: Violin plot of the absolute difference between observed and simulated aircraft samples of CO, in a simulation with GFAS (green) and in a simulation with MOPITT optimized biomass burning emissions (orange). Dashed lines inside each violin indicate the median and the two inner quartiles.




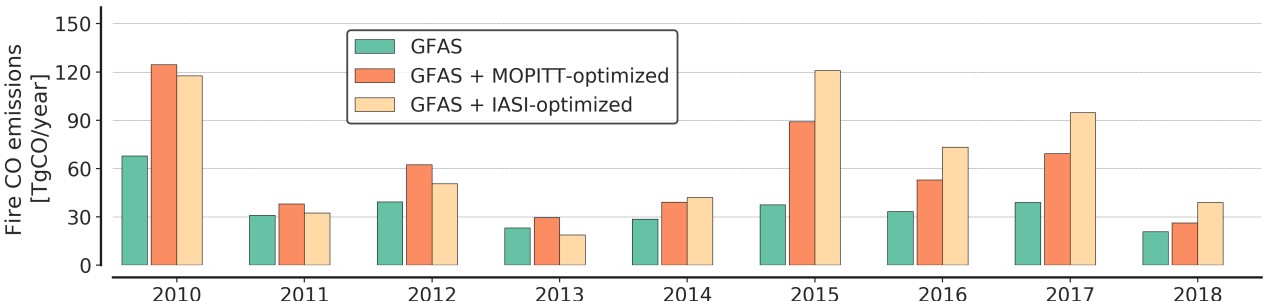

**Figure 7.** Total CO emissions from biomass burning summed over the $1°$ x $1°$ South-American zoom domain, and over the April $-$ December inversion period. Results for two sets of inversions are shown, which each started from the GFAS fire prior. The first is the default inversion that used MOPITT satellite data, the second used IASI satellite data.

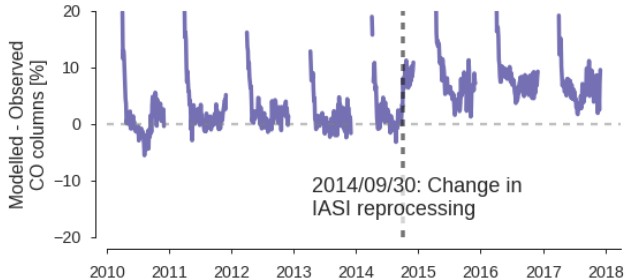

**Figure 8.** The difference between satellite-retrieved MOPITT CO columns and MOPITT CO columns sampled in a simulation that used CO emissions optimized with IASI-retrieved CO columns, over the $1°$ by $1°$ South-American zoom domain. The difference is quantified as a percentage of the satellite-retrieved CO columns. The date where IASI switches between two meteorological datasets (2014/09/30) is indicated, and on this date a jump in the difference between simulated and satellite-retrieved MOPITT CO columns occurs.

We have sampled MOPITT columns in simulations with IASI-optimized biomass burning emissions to investigate the tim-
ing of this jump (Fig. 8). Over 2010-2013, simulated MOPITT CO columns are in good agreement with those retrieved by MOPITT, which indicates consistency between the MOPITT and IASI records. However, in 2014 a jump occurs, after which simulated MOPITT columns become biased high, which is consistent with the difference in emissions. The onset of this bias of around 8% occurs instantaneously on 2014/09/30, as indicated in Fig. 8. Over the 2010–2018 period, several changes have been made to the IASI retrieval that can cause inconsistencies (e.g. Table 2 in Bouillon et al. (2020)), and a major update to
the processing algorithm that was implemented on 2014/09/30 apparently has a particularly large impact on the retrieved CO columns. Based on the coincidence of these two events, we consider the switch in IASI–MOPITT offset to be an artifact in the IASI data record.





We conclude that as long as the IASI retrieval does not use a consistent meteorological dataset, the retrievals before and after 2014/09/30 are best treated as two separate data records. Currently the IASI team is finalizing a full reprocessing of the

CO Metop-A record, using the ERA5 reanalysis as input for temperature profiles in order to generate a homogeneous record. The resulting consistent IASI product will provide better grounds for an uncertainty estimate of the driver satellite data, which is currently more difficult to perform. We do consider the relative consistency in interannual variability between IASI and MOPITT, excepting the 2014 break, evidence for robustness of interannual variability in CO emissions derived from either satellite product. The systematic difference between MOPITT and IASI is a measure for systematic uncertainty in the satellite

data and its impact on derived annual total fire CO emissions, which amounts to $10-30$ Tg/year for the inner zoom domain.

### 3.3.2  Integrated comparison to Zheng et al. (2019)

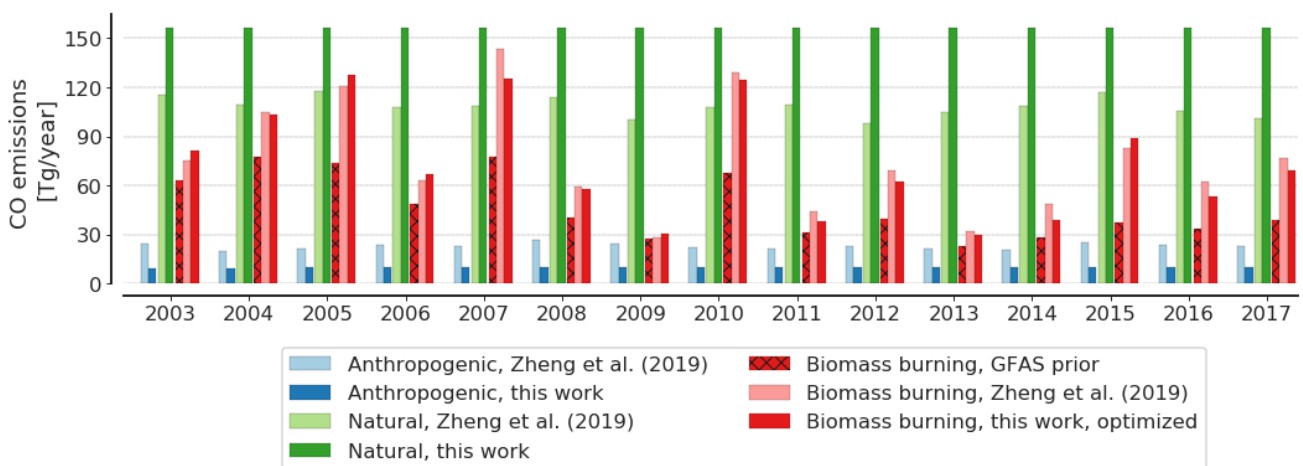

**Figure 9.** Emission totals for different source categories from this work, compared to emissions from Zheng et al. (2019). Total CO emissions are summed over the $1°$ x $1°$ South-American zoom domain, and over the April $-$ December inversion period. For bars corresponding to this work, emissions from our standard inversion are shown (i.e. GFAS fire emissions optimized with MOPITT-retrieved CO columns), as well as emissions from a global MOPITT inversion performed by Zheng et al. (2019). Emission categories from Zheng et al. (2019) were merged to obtain emission categories comparable to ours.

We have additionally compared our derived CO emissions to those derived by Zheng et al. (2019). Their emission estimate, covering 2000-2017, was derived in an inversion that also assimilated MOPITT-TIR CO column retrievals. However, that is the only shared aspect of our two inverse set-ups. Their inversion assimilated MOPITT CO column retrievals globally, which

will result in different boundary conditions for the Amazon domain than assimilation of surface observations. Additionally, their inversion was performed at a different spatial resolution ($3.75°$ longitude by $1.9°$ latitude, with 39 vertical layers), in a different transport model (LMDz-SACS (Pison et al., 2009)). As a prior for fire CO emissions, they used emissions from the Global Fire Emissions Database (GFED) 4.1s (van der Werf et al., 2017). Prior OH fields used in Zheng et al. (2019) were





the same as ours, but theirs were optimized with methyl chloroform surface observations. Additionally, they used biogenic

CO emissions from the MEGANv2.1 inventory, which, as discussed in Supp. 2.1, differ significantly from those used in our inversions. Further details on their inverse set-up are provided in Table 1 of Zheng et al. (2019). We limit our comparison to emissions derived in Inversion 1, as described in Zheng et al. (2019), in which satellite retrievals of formaldehyde and of methane were not assimilated.

We find that biomass burning emissions derived in Zheng et al. (2019) are comparable to ours, both in absolute magnitude and

in interannual variability (red bars in Figure 9), with an average annual total difference of $-3.0 \pm 6.7$ TgCO/year (one standard deviation). This difference is small compared to interannual variability. Notably, their emission estimates of CO from biomass burning are also systematically higher than those from GFAS and FINN. Different from our inversions, the MEGANv2.1 inventory for biogenic emissions includes interannual variability, and these emissions are significantly lower than the biogenic emissions that we have used (see also Fig. S3). The excellent agreement between these two largely independent estimates

provides much confidence to the final emission estimates.

### 3.3.3 Other sensitivities in the inverse system

We have additionally explored the sensitivity of derived emissions to other, individual components of the inverse system, which are presented in detail in Supp. 2. Here, we briefly summarize the main conclusions from these sensitivity tests. We find that natural production of CO from non-methane volatile organic compounds (NMVOCs) is the largest sensitivity in our inverse

system (Sup. 2.1), with an associated systematic uncertainty in derived fire CO emissions of 23–27 Tg/year. The associated uncertainty in interannual variability of fire CO emissions at 10–15 Tg/year. We additionally find a large sensitivity to the OH sink of CO, but we attribute this mostly to unrealistically low OH values in the fields from the CAMS reanalysis that we use for the sensitivity test (Sup. 2.2). Finally, we find that if we reduce the error on NOAA surface observations, we retrieve a better posterior match with these data, without changing the derived fire emissions. This result indicates limited sensitivity to

boundary conditions as determined by surface observations that are sampled mostly outside the domain in which we assimilate satellite data.

Overall, we conservatively estimate the uncertainty in the interannual variability of the MOPITT-derived CO emissions of biomass burning at 10–15 TgCO/year, and the systematic uncertainty at 30 TgCO/year, which is dominated by production from NMVOCs. We consider this uncertainty estimate conservative, since the integrated comparison with Zheng et al. (2019)

suggests a lower uncertainty of 7 TgCO/year (Sect. 3.3.2). A small uncertainty is somewhat intuitive, since fire emissions are uniquely sharp in location and timing, and this signal is well captured in the MOPITT data. Therefore, fire emissions are only partly interchangeable with other, typically more diffuse budget components.

### 3.4 Discussion

The robustness of derived emissions signifies the detail provided by the MOPITT-TIR product. In addition to the TIR product,

MOPITT also provides a NIR product and a combined NIR-TIR product. The NIR product has relatively higher vertical sensitivity near the surface. Due to its range of spectral bands, MOPITT data can be used to separately constrain upper and





lower tropospheric CO (Deeter et al., 2018). Here, we have limited our analysis to the TIR product, which already provides strong constraints on CO fire emissions. Nechita-Banda et al. (2018) showed that over Indonesia the TM5-4DVAR inverse system produces similar fire emissions when MOPITT-TIR or NIR-TIR are assimilated. Peiro et al. (2022) performed a global CO inversion with MOPITT NIR-TIR data and found South American fire emissions that were typically lower than ours. Therefore, in future work, it would be interesting to investigate the added value of NIR in the NIR-TIR product in more detail. Additionally, we use MOPITT version 8, but recently the newer MOPITTv9 version has become available (Deeter et al., 2021). Importantly, the changes made in the retrieval result in significantly improved coverage over areas with high aerosol concentrations, which are also emitted in fires. Clearly, improved coverage near fires will strongly benefit our inverse analysis, and future work can benefit from these improvements.

Our fire emission estimates rely strongly on the quality of the MOPITT CO retrieval. The MOPITT CO data have been validated extensively (Deeter et al., 2016, 2019), and in this work we again present a good agreement with independent aircraft profiles. Other CO satellite products are available, which can complement a MOPITT-based analysis. Firstly, we presented a comparison to inversions that use IASI instead of MOPITT CO data (Section 3.3.1). This comparison reveals inconsistencies in the reprocessing used for IASI, but the interannual variability derived from the two products within the 2010–2013 and 2015–2018 periods is similar. A consistently processed IASI product, which is something that the IASI team is finalizing, will help better assess the MOPITT-derived emissions. Additionally, an expanding fleet of satellite instruments is becoming available, which monitor atmospheric composition with increasing detail. This can help with cross-validation, and new satellites, such as TROPOMI (Borsdorff et al., 2018), also have higher spatial resolution, providing a potential step forward in the level of detail on which fire emissions can be inferred (van der Velde et al., 2021). We do note that in this work we draw value from the long-term availability and consistency of the MOPITT product, which is something that other products currently cannot compete with.

An important application of top-down estimates of CO fire emissions is to propagate these to $CO_2$ fire emissions, so that the Amazon carbon balance can be better constrained. In previous work, this has been done by directly applying $CO:CO_2$ emission factors from bottom-up inventories to the updated CO emissions (van der Laan-Luijkx et al., 2015; Peiro et al., 2022). In our case, this would mean that the $CO_2$ fire emissions over the Amazon in the GFAS and FINN inventories would be scaled up, since we find that CO emissions in these inventories are underestimated. Whether this approach is appropriate strongly depends on the driver of the underestimate we have found. Scaling up the $CO_2$ emissions based on our CO inversion is appropriate if the underestimate of CO fire emissions is related to missed understory fires (Alencar et al., 2004), or to an underestimate in carbon stock. However, if the underestimate is related to errors in the CO emission factors (van Leeuwen et al., 2013), then the total carbon emissions reported in the bottom-up inventories could still be accurate. As a first indication, we find that the underestimate in GFAS is largest over savanna and shrubland regions, which makes it less likely that understory fires are a dominant driver. A recent study has shown that a combined analysis of satellite data of CO and $NO_x$ can provide top-down constraints on combustion efficiency and emission factors (van der Velde et al., 2021). Additionally, a burned-area analysis of high-resolution Sentinel-2 data over Africa concluded that missed small fires in GFED4s might result in an underestimate of fire carbon emissions of 31% (Ramo et al., 2021). The updated version of the FINN fire inventory (v2.5; (Wiedinmyer et al.,



2022)) also increases fire CO emissions in the Amazon basin by 102% (averaged over the 2003–2018 period) compared to the version used in this work (v1.5; (Wiedinmyer et al., 2011)). These new developments show that there is perspective on reconciling our top-down estimates with bottom-up efforts, which can be further informed by the spatio-temporal patterns of
the higher emissions we derive.

An operational framework that estimates CO fire emissions based on satellite-retrieved CO columns and prior information (e.g. FRP) can provide unique and timely information about regional variability in fires. The Copernicus Atmospheric Monitoring Service (CAMS) already provides an operational data assimilation framework in which, among other data products, MOPITT-TIR CO column retrievals are assimilated (Flemming et al., 2017). We identify two aspects in which the CAMS
analysis can be improved. Most importantly, in the CAMS system the atmospheric abundance of CO is optimized, instead of CO emissions. We suggest that a next development of a CAMS-like system should consider emission optimization for a more physically realistic end-product that can be used to inform on variations in sources, such as in Miyazaki et al. (2020). Secondly, we show that the CAMS OH fields produced in full-chemistry simulations are very low over the Amazon, which other studies have indicated is due to incomplete $NO_X$ sources (Wells et al., 2020), or incomplete chemical mechanisms (e.g. Lelieveld
et al., 2008; Taraborrelli et al., 2012). Such an underestimate in OH can mask the GFAS underestimate that we find here, which has important implications for the interpretation of fire emissions. Operational CO emissions can provide a rich proxy for fire variability and deforestation. Moreover, the immediate context provided by a long-term, consistently derived timeseries of CO emissions is highly valuable for interpretation of recent fire events. This would be timely, considering that recent changes in the natural and political climate surrounding the Amazon (INPE, 2020; Silva Junior et al., 2020; Fonseca et al., 2019) necessitate
active fire monitoring by as many independent proxies as possible.

## 4   Conclusions

In this study, we present a 2003–2018 timeseries of fire CO emissions in the Amazon domain. Importantly, our derived emissions are robust against the exchange of prior distributions of fires from several bottom-up efforts, even at the scale of specific Brazilian states and land-use types. Moreover, we find that simulations with optimized fire emissions better reproduce indepen-
dent aircraft observations than simulations with prior emissions from the GFAS or FINN inventories. The largest uncertainty in the inverse system derives from uncertainty in CO production from non-methane hydrocarbons (NMHCs; see Supp. 2.1), and we conservatively estimate a combined uncertainty in interannual variations of basin-wide emissions of 10–15 Tg/year (Sect. 3.3.3). We see this robust method to detect, attribute and quantify fire CO emissions in the Amazon as a valuable addition to the palette of existing fire monitoring methods for the region.
Variations in CO emissions over our 2003–2018 study period are a combination of strong interannual variations and a long-term decrease, mostly between 2003–2012. Interannual variations are closely correlated with variations in the Amazonian water balance, evident from a strong link with soil moisture even at state-level. In contrast, the long-term decline in CO emissions over the 2003-2012 period mirrors a decrease in deforestation rates, especially so in forested regions. These results emphasize the positive effect of deforestation abatement policies, as well as the potential impact of increased drought frequency in a



changing climate. As such, sustained efforts to reduce deforestation can reduce the impact of climate change on fire risk, while a return to deforestation rates of the early 2000's in a drier climate likely results in enhanced fire risks.

*Author contributions.* WP, SN, MK and IL designed the research. SN wrote the manuscript with major input from WP, MK and IL, and further contributions from all co-authors. The basis of the TM5-4DVAR set-up used in this study was developed by SB. SN performed the TM5 inversions and analysed the results. WP and MK supervised the research. All authors discussed the results and contributed to the final 410 manuscript.

*Competing interests.* The authors declare that they have no conflict of interest.

*Acknowledgements.* This work was carried out on the Dutch National e-Infrastructure with the support of SURF Cooperative. This work was funded through the Netherlands Organisation for Scientific Research (NWO), project number 824.15.002. This project has additionally received funding from the European Research Council (ERC) under the European Union's Horizon 2020 research and innovation programme 415 under grant agreement No 742798. W.P. and G.K. received funding from the European Research Council under grant no. 649087: ASICA (Airborne Stable Isotopes of Carbon from the Amazon). I.T.L. received funding from the Dutch Research Council (NWO) (Veni grant 016.Veni.171.095). The development of the CO 4D-Var system was partly funded via NASA Carbon Monitoring System program Interagency Agreement NNH16AD06I. IASI is a joint mission of EUMETSAT and the Centre National d'Etudes Spatiales (CNES, France). The authors acknowledge the AERIS data infrastructure for providing access to the IASI data in this study, and Maya George and Cathy Clerbaux for 420 scientific discussions. The specific TM5-4DVAR set-up used in this work was in large part developed by Narcisa Banda.

## Data availability

The optimized CO emissions that result from the reference GFAS inversions are available online at DOI:10.6084/m9.figshare.14294492. Additional data is available on request. The base TM5-4DVAR code is open-source and available on: https://sourceforge.net/projects/tm5/. The specific code used for our inversions is available on request. FINN fire 425 emission data is available on: https://www.acom.ucar.edu/Data/fire/. GFAS fire emission data is available on: https://apps.ecmwf.int/datasets/data/cams-gfas/.



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
