# Peer review of "Sixteen years of MOPITT satellite data strongly constrain Amazon CO fire emissions"

_EGUsphere, 2022_

## Author Comment (AC1)

We thank the reviewer for the kind words, and for taking the time to provide feedback on the manuscript, which has helped improved its quality further.

Line-by-Line Comments:

Line 2. By how much do these estimates vary by?

We choose not to become more quantitative here, as that would require references in the abstract. There are several examples that indicate how there might be large uncertainties in carbon fire emissions. A good example is given in Ramo et al. (2021), which used a high-resolution burned-area dataset, derived from Sentinel-2, to show that carbon emissions in GFED4s might be underestimated by up to 31% over Africa, since the coarser-resolution MODIS burned-area product used by GFED misses small fires. More generically, uncertainties in biomass burnt translate to uncertain carbon emissions. Clearly, there is a role for top-down studies such as ours to better understand and reduce uncertainties.

Line 8. A fourfold increase over wet years?

Yes, we have made this explicit now.

Line 5. I'm confused whether "3-daily" refers to 3 times per day or every 3 days. Please make this clear here.

We have adjusted phrasing to indicate we mean every 3 days (here and throughout the manuscript).

Line 24-25. "albedo changes" needs to be corrected to burned area or surface reflectance. GFED (van der Werf et al., 2017) is primarily based on the MODIS burned area product. Burned area classification is derived from changes in surface reflectance. The references the authors list refer to quantification of fire emissions rather than pure monitoring of fires. For the latter, the authors should cite papers that describe the active fire and burned area products, such as Giglio et al. (2016) and Giglio et al. (2018). I'm not sure what the authors mean by the products being "partly related." Related in what way? Being able to serve as the basis for emissions estimates?

We have rephrased to separate between fire monitoring and emission estimation (L23-26). "Partly related" referred to the fact that bottom-up emission estimates sometimes share assumptions (e.g., landcover classification or emission factors), but we have removed this phrase for clarification.

Line 71. Please state the spatial resolution of the ERA-Interim reanalysis product.

We have provided this now in L74-75.

Line 77. The authors say GFASv1.2 is provided at 0.5° spatial resolution here, but GFASv1.2 is provided at 0.1° spatial resolution.

Corrected.

Line 88. "three-day" and "three-daily" are used interchangeably. Stick with one or the other.

We have adjusted to three-day total for clarity and consistency.

Line 90. Why a 0.03 Tg threshold specifically? Is this a statistical cutoff?

This is a pragmatically chosen number that we found includes a large area still (24% of the inner domain area), but not all grid cells. In this way, it is somewhat arbitrary, but rather than the exact value chosen here, the important conclusion is that we can start from a completely flat prior in time that contains very little spatial information, and still retrieve results close to our GFAS- or FINN-based estimates. This insensitivity to the prior is rare in inverse modeling studies. We do not see how a slightly different cut-off choice would impact that conclusion.

Line 95. Why were the GFAS emissions outside the domain not averaged? How much does the interannual variability of the emissions outside the domain influence the results?

We wanted to determine how uniquely MOPITT data determines the fire emissions in our domain of interest, i.e., which part of the posterior solution comes from the prior, and which part from the satellite data. In the outer zoom domain (i.e., the global domain) we are reliant on the variability from bottom-up inventories, as we only assimilate surface observations. Moreover, we are not interested in the posterior fire emissions we derive there.
We considered the middle zoom domain also part of the boundary conditions, whereas we wanted to test the influence of the fire prior. However, this is a design choice, and we could equally well have run this sensitivity test with a flat prior in the middle domain. Given that we find strong constraints from MOPITT in the inner domain, we consider that we would similarly find strong constraints in the middle domain, and thus that boundary conditions would be little affected, but we have not performed this test.
In the FINN inversion, the fire emissions in all domains are of course changed relative to the GFAS/CLIM inversions, and here we find similar emission estimates as in the GFAS inversion. Additionally, we have tested the influence to boundary conditions by adjusting the error on surface observations (Section 3.3.3 and Supplement 1), which resulted in little change in emissions. This indicates that most emission information comes from inside the inner domain.

Line 115. Please explain why a factor of the square root of 50 is chosen.

The square root 50 factor was pragmatically found to increase the observational cost function of an inversion that assimilates both surface and satellite data by a factor two relative to an inversion that only assimilates surface data (Hooghiemstra et al., 2012). I.e., this indicates that satellite data and surface observations have equal weight in the inversion. Without this factor, the large number of satellite data dominate the observational cost function, and the surface observations (especially near the Amazon domain) are not fitted within one-sigma.
In hindsight, it might have been appropriate to the error even more, since we find that in the base inversions surface observations near the Amazon domain are not always well-reproduced, indicating that the satellite data still have too much weight in the inversion. However, when we reduce the error on surface observations by a factor 10 (giving them more

weight in the inversion), we match the surface observations better, we still retrieve a good match with MOPITT, and we find similar posterior CO fire emissions. Therefore, it doesn't seem that the posterior fire emissions are very sensitive to this factor.

Since this factor is so specific to the inversion set-up, we do not consider it worthwhile to include a recommendation in the manuscript.

Line 163-171. The authors should quantify the interannual variability, e.g. standard deviation. In general, the authors should be more quantitative in describing their results.

We agree with the reviewer, and we have substantiated the manuscript section with more quantitative descriptions of the results.

Figure 4. The black circles in 4a shows how much CO is added to the fires, but a scale/legend is needed here.

We have added a legend to Fig. 4a.

Figure 5. It's hard to see the purple "Observed" line in the top panel. Since the difference between "Observed" and "Simulated" are shown in the bottom panel, just showing the "Observed" line in the top panel might be a better approach.

We now only show the "observed" line in the top panel.

**References**

- Hooghiemstra, P. B., et al. "Interannual variability of carbon monoxide emission estimates over South America from 2006 to 2010." *Journal of Geophysical Research: Atmospheres* 117.D15 (2012).
- Ramo, Ruben, et al. "African burned area and fire carbon emissions are strongly impacted by small fires undetected by coarse resolution satellite data." *Proceedings of the National Academy of Sciences* 118.9 (2021): e2011160118.

---

## Author Comment (AC2)

We thank the reviewer for the kind words, and for taking the time to provide feedback on the manuscript, which has helped improved its quality further.

*Line 5: Why don't you use 2003-2021 in your analysis, especially since you say on Line 12 that 2019-2021 are interesting years?*

Indeed, 2019-2021 are interesting years with large Amazon fires, and it would be interesting to investigate these years with our model framework. However, expanding to more recent years is non-trivial in part because ERA-Interim (used in this study) was replaced by ERA-5. ERA-5 was not available when this project started, and ERA-Interim is not available for 2019-2021. Therefore, for consistent results in recent years, we would have to redo the inversions for the whole timeseries with ERA-5. This does not fit within the scope of the project in which this research was performed.
We do consider investigation of 2019-2021 an interesting target for follow-up research. Indeed, in our research group an investigation of a coupled CO-$CO_2$ system is underway with more focus on recent years.

*Line 45: In this paragraph, you are trying to say what's new about your work as compared to other studies in the literature. Your topic sentence seems to address the novelty of your work (i.e., data assimilation), but then you say in the next sentence that others have done this as well. That is, you only give a few sentences about the previous work which only raise questions about the novelty of your work. Your new aspect seems to be that you are looking at a longer time period than in other studies. If this is true, this is a weak justification unless there is something unique about the additional years. MOPITT data have been around for a very long time and many studies have been done, so I strongly recommend that you expand discussion on these previous studies and clearly articulate how your work is new.*
*Note: Upon reading further, I see that you devote Section 3.3.2 to Zheng et al. (2019). This makes it even more important for you to clearly differentiate between Zheng et al. and your work in the introduction.*

What makes this study unique is the combination of the use of most of the long MOPITT timeseries, data assimilation, and a focused and rigorous analysis of the system and its results over the Amazon. We are not aware of any previous work that addressed these three together.
The focus and depth in our work add context to global studies such as Zheng et al. (2019). Since we find that our estimates are mostly robust, it is unsurprising that our estimates align with theirs, but this was not obvious a priori. Moreover, the follow-up analysis, e.g., at state-level, is novel, as is the robustness of our estimates at this level.
The data assimilation makes our analysis more quantitative than e.g., Aragao et al. (2018), who also did a focused analysis of the MOPITT timeseries over the Amazon, but without data assimilation or a transport model, which makes attribution of MOPITT CO anomalies more uncertain.
Finally, data assimilation studies with in-depth focus on the Amazon have generally investigated shorter time periods (e.g., Hooghiemstra et al., 2012; Luijkx et al., 2015).
We consider that our work is timely and important precisely because these components have received much attention individually (or in two's), but never together.

Finally, since in-situ observations in the Amazon are rare, the independent validation of our inversion results with aircraft profiles is highly valuable. Since aircraft observations have a different vertical sensitivity than satellite data, having this incorporated in our transport model analysis makes such a comparison more quantitative than a direct comparison as in e.g. Deeter et al. (2018).

We agree that this was not clearly explained in the manuscript, and we have rephrased this part (L48-64).

*Line 110 & Line 151: Why not assimilate satellite retrievals over the whole globe? Is it simply because such an inversion would be computationally expensive as suggested in the paragraph beginning on Line 146? It seems that it would make more sense to do the assimilation for the whole globe so that your background CO will be more realistic, especially since your OH and CO production from methane/VOC oxidation are both static (Line 102-108). What are the implications for your study by not accounting for the background trend in CO over your study period?*

Our approach does account for the background trend in CO through assimilation of surface observations and optimization of total CO emissions globally. Indeed, the advantage of our approach is partly computational expense. Additionally, setting up a correct inversion and interpreting the result becomes much more complex when satellite data are assimilated globally, since in the global CO MOPITT data there are many features that we are not interested in here, but that would influence the inversion. For example, it is unclear what the detail provided by satellite data over Europe would add to our Amazon-focused analysis, but once we include such data, we need to properly represent it in the model. We do include satellite data outside our domain of interest (i.e., over the middle zoom domain), which, together with the surface data, provide boundary conditions to the inner domain. In the end, this is a design choice, and, though there are advantages to assimilating satellite data globally, it is not obvious that these would benefit our research objective much.

*Minor Comments*

*Line 5: "3-daily" is unclear. Replace "3-daily" with "3-day average". I think that's what you mean.*

We have implemented the suggestion, and now no longer use the ambiguous 3-daily anywhere else in the manuscript either.

*Line 62: You said in the introduction that the model framework is comparable to other setups. Please clarify if the other studies used the same setup or some of the same components.*

It depends on which study. In the introduction (L50 in the pre-print) the comparison between e.g., our work and Zheng et al. (2019) was drawn to emphasize that we both use a transport model inversion to estimate emissions, whereas other work (e.g., Aragao et al. (2018)) interpreted MOPITT CO columns directly. In that comparison, the transport model, meteorological driver data, prior emission and loss fields and inversion framework are all different, as is also described in Section 3.3.2.

In the Methods, we draw the comparison to Luijkx et al. (2015), in which a TM5-4DVAR inversion was performed to estimate CO fire emissions in the Amazon. In that case, the comparison indicates that we use the same transport model, meteorological data and inversion framework, but choices made within that framework are often different. For example, the zoom domains differed slightly, prior emission fields were different, and in that work NMHC emissions of CO were co-optimized on a monthly timescale. Finally, the reference inversions assimilated IASI instead of MOPITT.

We consider that we provide the relevant comparisons where necessary already, and as both this work and previous work describe the inversion set-up, interested parties can look into the details themselves.

*Line 125: Again, why just assimilate in situ observations when you have satellite retrievals?*

We refer to our previous answer.

*Line 316: You didn't mention how MOPITT's averaging kernels may introduce uncertainty in the inversion.*

We are not entirely sure what is meant with this comment: that there is uncertainty in the vertical sensitivity of MOPITT, or that the vertical sensitivity of MOPITT limits what we see? Either way, we perform inversions with IASI, which has a different vertical sensitivity than MOPITT, and we present an independent validation with aircraft data which also have their own vertical sensitivity. Finally, we mention in the discussion how in future work it would be interesting to explore synergy between MOPITT TIR and NIR CO, as well as TROPOMI SWIR, which all have their own vertical sensitivities. With this, we consider that we cover this topic sufficiently.

**References**

- Zheng, Bo, et al. "Global atmospheric carbon monoxide budget 2000–2017 inferred from multi-species atmospheric inversions." *Earth System Science Data* 11.3 (2019): 1411-1436.
- Hooghiemstra, P. B., et al. "Interannual variability of carbon monoxide emission estimates over South America from 2006 to 2010." *Journal of Geophysical Research: Atmospheres* 117.D15 (2012).
- van der Laan-Luijkx, Ingrid Theodora, et al. "Response of the Amazon carbon balance to the 2010 drought derived with CarbonTracker South America." *Global Biogeochemical Cycles* 29.7 (2015): 1092-1108.
- Aragão, Luiz EOC, et al. "21st Century drought-related fires counteract the decline of Amazon deforestation carbon emissions." *Nature communications* 9.1 (2018): 1-12.